# Automatic Recognition of Fish Behavior with a Fusion of RGB and Optical Flow Data Based on Deep Learning

**DOI:** 10.3390/ani11102774

**Published:** 2021-09-23

**Authors:** Guangxu Wang, Akhter Muhammad, Chang Liu, Ling Du, Daoliang Li

**Affiliations:** 1National Innovation Center for Digital Fishery, China Agricultural University, Beijing 100083, China; gxwangmail@163.com (G.W.); muhammadakhter58@gmail.com (A.M.); chang_liu@cau.edu.cn (C.L.); duling0615@163.com (L.D.); 2College of Information and Electrical Engineering, China Agricultural University, Beijing 100083, China; 3Beijing Engineering and Technology Research Centre for Internet of Things in Agriculture, Beijing 100083, China; 4Key Laboratory of Agricultural Information Acquisition Technology, Ministry of Agriculture, Beijing 100083, China

**Keywords:** deep learning, fish behavior, image processing, video sequences

## Abstract

**Simple Summary:**

Animal behaviors are critical for survival, which is expressed over a long period of time. The emergence of computer vision and deep learning technologies creates new possibilities for understanding the biological basis of these behaviors and accurately quantifying behaviors, which contributes to attaining high production efficiency and precise management in precision farming. Here, we demonstrate that a dual-stream 3D convolutional neural network with RGB and optical flow video clips as input can be used to classify behavior states of fish schools. The FlowNet2 based on deep learning, combined with a 3D convolutional neural network, was first applied to identify fish behavior. Additionally, the results indicate that the proposed non-invasive recognition method can quickly, accurately, and automatically identify fish behaviors across hundreds of hours of video.

**Abstract:**

The rapid and precise recognition of fish behavior is critical in perceiving health and welfare by allowing farmers to make informed management decisions on recirculating aquaculture systems while reducing labor. The conventional recognition methods are to obtain movement information by implanting sensors on the skin or in the body of the fish, which can affect the normal behavior and welfare of the fish. We present a novel nondestructive method with spatiotemporal and motion information based on deep learning for real-time recognition of fish schools’ behavior. In this work, a dual-stream 3D convolutional neural network (DSC3D) was proposed for the recognition of five behavior states of fish schools, including feeding, hypoxia, hypothermia, frightening and normal behavior. This DSC3D combines spatiotemporal features and motion features by using FlowNet2 and 3D convolutional neural networks and shows significant results suitable for industrial applications in automatic monitoring of fish behavior, with an average accuracy rate of 95.79%. The model evaluation results on the test dataset further demonstrated that our proposed method could be used as an effective tool for the intelligent perception of fish health status.

## 1. Introduction

Due to rapid growth in the global population, the demands for aquatic products have increased rapidly for the last two decades [1]. In response to this rapid increase in demand for aquatic products, in many countries, fishery products supply has been dominated by farming fish. However, this industry worldwide is facing huge losses due to a lack of adequate fish health and welfare monitoring and management practices. Animals typically respond to and interact with the environment using a series of rich behaviors. The ability to recognize behavior activities from video data creates a need for tools that automatically quantifies behavior and simultaneously provides a deeper understanding of the relationship between behavior and the external environment [2]. Although significant progress has been made, the challenge of fish behavior recognition is still far from being solved due to the lack of research on the specific posture of fish. Automated recognition and classification of fish behavior will aid in this task, which is considered an efficient method for long-term monitoring and assessment of fish health and welfare [3].

One common approach is to manually observe behavior videos over a long period of time, which is labor intensive, and inefficient. Researchers have developed and designed a large number of sensors and algorithms to automatically and accurately identify fish behavior. The biological telemetry method has excellent stability in identifying the behavior of single and fewer fishes, which has been used for animal welfare monitoring and behavior recognition, such as environmental stress behavior and swimming behavior [4]. This integrated micro-sensor has the advantage of simultaneously measuring multiple motion parameters such as movement speed, acceleration, and turning angle. However, the sensor needs to be implanted into the skin or body of the fish, which affects the behavior of the fish and also damages fish welfare. Therefore, hydrophone and sonar technologies based on active acoustic were used for fish monitoring and tracking, behavior monitoring, and biomass estimation in the deep sea [5,6]. The acoustic imaging method has the advantage of capturing high-resolution video even in a muddy and dark underwater environment, but this method relies on expensive acoustic equipment.

The emergence of computer vision technology in aquaculture has opened new possibilities to observe fish behavior effectively and efficiently. Traditional computer vision techniques for identifying fish behavior relied on extracting important features such as motion patterns, clustering index, and trajectory. Zhou et al. (2017) precisely quantified the feeding behavior of fish by calculating the clustering index of fish in the near-infrared image with an accuracy of 94.5%, providing an effective method for analyzing the feeding behavior of fish [7]. The video tracking algorithm creates the possibility of automatically identifying fish behavior, and has been widely used to measure the swimming parameters (speed, acceleration, and rotation angle) of fish. A machine vision system with a monocular camera was used to evaluate the swimming performance of fish under different water temperatures [8]. In addition, 3D imaging systems based on stereo vision have been applied to fish detection and tracking [9]. In particular, 3D images can provide more spatial information to overcome the problem of occlusion during movement, and the tracking accuracy rate reached 95%. In addition, the stress level of fish has been assessed by monitoring the abnormal trajectory of the fish [10,11]. This method is helpful for the early diagnosis of fish disease and optimizes management practices in aquaculture.

Deep learning methods have revolutionized our ability to automatically analyze videos, which have been used for live fish detection [12], species classification [13], biomass estimation [6], behavior analysis [14], and water quality monitoring [15]. The advantage of deep learning is that it can automatically learn to extract image features and shows excellent performance in recognizing sequential activities. Zhou et al. (2019) proposed an automatic classification method of fish feeding intensity based on a convolutional neural network (CNN) and machine vision [16]. The classification accuracy of this method reached 90% for fish appetite (strong, medium, and weak). In recent years, researchers have discovered that it is essential to combine spatial and temporal features to identify fish behavior. A two-stream recurrent network composed of a spatial network, a 3D convolutional network, and a long short-term memory network (LSTM) was used for salmon behavior recognition [17], which integrates spatial and temporal information, and the accuracy of behavior prediction was 80%. Aiming to monitor the local abnormal behaviors of fish in intensive farming, a method based on an improved motion influence diagram and recurrent neural network (RNN) was proposed [18]. The detection, location, and recognition accuracy rates of this method were 98.91%, 91.67%, and 89.89%, respectively.

Although deep learning methods have achieved high accuracy in target detection and tracking, fish behavior recognition methods need to be further improved to adapt to the complex underwater environment and rapid target movement. Fish behavior usually reflects the stimulus of the external environment through changes in swimming behavior. For two different types of behaviors, a video sequence may exhibit a similar activity, which leads to a decrease in recognition accuracy. For example, both feeding behavior and frightening behavior include the rapid movement of fish after being stimulated. In addition, one type of behavior contains a variety of different activity states. For example, fish exhibit the behavior of sinking on the side of the water tank when experiencing hypoxia but also actively gather at the water injection port. Therefore, subtle changes between different behavior types and multiple actions of the same behavior are the main challenges for the application of deep learning in fish behavior recognition.

To meet the above challenges, in this paper, a DSC3D model based on temporal and spatial feature information is proposed to recognize the behavior states of the fish school from the collected video data. The main contributions of our work are as follows: (1) a novel method based on deep learning is used to recognize fish activities; (2) proposed framework recognizes the five behaviors of fish through the feature fusion of RGB images and optical flow images; (3) proposed framework shows excellent performance and satisfactory results. Our proposed method provides an effective strategy for automatically identifying fish behavior from video sequences in real time, laying the foundation for intelligent perception in aquaculture.

## 2. Materials and Methods

### 2.1. Fish

In this experiment, we used *Oplegnathus punctatus* as an experimental subjects, which is a farmed fish located in Yantai, Shandong Province, China. Before the experiment, the fish had lived in the recirculating aquaculture system for 6 months. All fish tanks were equipped with automatic bait feeding machines, which were set to feed bait 3 times a day.

### 2.2. Experimental Setup

The experimental system consisted of a recirculating aquaculture system and a computer vision system, as shown in Figure 1. The fish tank was a cylindrical shape with a diameter of 5m and a height of 1m. The cultured water was supplied from the nearby seawater that was processed and filtered by the factory. Water quality parameters including dissolved oxygen (DO) at (5 ± 0.6) mg/L, total ammonia nitrogen (TAN) < 0.5 mg/L, water temperature (15 ± 2) °C, pH(7.2 ± 0.5) can be fine-tuned to obtain the behavior changes of fish as for the machine vision system. A high-definition digital camera (Hikvision DS-2CD2T87E(D)WD-L) with a frame rate of 25 fps (1920 × 1080) was deployed on a tripod with a height of about 2 m to captured the video data, as shown in Figure 1. The deep learning model was implemented by writing code based on the Pytorch framework on the Linux Ubuntu 18.04.3 LTS environment. It was trained with an SGD optimizer on a PC with 4 Nvidia RTX 2080 Ti. 

### 2.3. Data Acquisition

To better experiment and verify the effectiveness of our proposed method, we collected 24 h of video daily for approximately 3 months, resulting in a collection of behavior videos of different sizes, moments, and numbers of fish. All individuals were captured through the top camera, as the behavior of fish is usually characterized by group activities. The video was stored in the memory card of the network camera and regularly downloaded from the memory card to our computer. Our research was focused on recording the changes in fish behavior at different stages within 24 h. In addition to some common behaviors such as feeding behaviors, other behaviors that interacted with the environment were also collected, such as hypothermia, hypoxia, and startle stimulation of fish.

With the help of expert experience and knowledge, these behaviors were classified into feeding behaviors, hypoxic behaviors, hypothermia behaviors, frightening behaviors, and normal behaviors, as shown in Table 1. Our research aims to identify these five behaviors, which provide intelligent monitoring of fish health by identifying the active state of the fish when it is stimulated by food, sound, light, etc. In contrast, the hypoxia and hypothermia behaviors of the fish school were inactive, implying risks that may occur during the breeding process.

### 2.4. Data Annotations

To perform our dataset more diverse and representative, we selected as many different types of behaviors as possible from the videos. A trained observer manually annotated five types of actions for 2 weeks. In total, 1066 video clips were annotated. Automated analysis of these video clips was requisite for the full characterization of fish behavior in this limited dataset. As a result, we further manually classified the behavior on the keyframes in order to obtain the real situation, as shown in Figure 2.

Due to the large displacement of fish and rapid changes in body posture, accurate understanding of fish behavior on keyframes is the main problem in the process of dataset labeling. We invited aquaculture experts to label our behavioral data. In the end, we completed the extraction and annotation of keyframes for behavioral videos at a frame rate of 4FPS. Since we collected the data on the actual farm, occlusions, and reflections of lights on the water surface were also taken into consideration.

### 2.5. Behavior Dataset Statistics

The number of videos for each behavior in our dataset is shown in Table 2. We manually cropped video clips, each containing an average of 8 s of behavior. The demand for a large amount of data used for deep learning was solved by data enhancement. Data enhancement was achieved by rotating and flipping the image, and the number of video clips was expanded by 2 times. In the end, our dataset had a total of 3198 videos. Among them, 70% of the data was used for model training, and 30% of the data was used for model evaluation. Since the behavior of fish after domestication usually exhibits group characteristics, our dataset was mainly used for the behavior recognition of the fish school. A full dataset was established through a larger-scale investigation of fish behavior and communication with fishery experts to overcome the lack of knowledge about fish behavioral understanding.

## 3. Fish Behavior Recognition

The purpose of our research is to recognize and understand fish behavior through intelligent methods. Inspired by human behavior recognition, we attempted to apply deep learning methods to the automatic recognition of fish behavior. In this section, we introduce the proposed network architecture, and then we present image preprocessing, spatiotemporal feature extraction, and image fusion schemes.

### 3.1. Overview of the Network Framework

With the rapid development of deep learning, the C3D network has shown excellent performance in human temporal action recognition [19,20]. Inspired by the spatiotemporal network method based on deep learning, we improved the dual-stream convolutional neural network to realize the behavior recognition of the fish school. In this work, we proposed the DSC3D network to extract spatiotemporal features and motion features from RGB video and optical flow video, respectively. The spatiotemporal features describe the spatial distribution, appearance, and fish movement in consecutive frames. The motion features specifically representing the direction and speed of the fish swimming. Moreover, our network completes the fusion of spatiotemporal features and motion features.

The network framework of this article is shown in Figure 3. Our behavior recognition process was based on video analysis and included the following three parts: (1) image preprocessing and motion detection: we effectively removed water surface noise through image preprocessing methods and applied the FlowNet2 method based on deep learning for motion detection; (2) feature extraction based on a spatiotemporal network: our framework used improved C3D as the backbone network to extract spatiotemporal features and motion features from fish behavior videos; (3) feature fusion: a feature map combined spatiotemporal features and motion features for behavior recognition.

### 3.2. Image Preprocessing and Motion Detection

Image preprocessing is an essential process to eliminate image noise and improve the accuracy of feature extraction and image segmentation. Image preprocessing methods include grayscale, geometric transformation, and image enhancement. As explained in Section 2.3, we processed the dataset through image enhancement methods. As a requirement of the C3D model, we adjusted the frame rate of all input videos to 16 frames and the image size to 112 × 112.

The imaging quality underwater is easily affected by water surface fluctuations and light refraction compared with human behavior videos. To eliminate the image flare formed by the reflection of the water surface, the contour of the fish body was segmented by the grayscale method. Then, to avoid interference unrelated to fish activity, we used the Gaussian kernel convolution filtering method to remove the noise caused by water surface fluctuations in the image. We obtained a 3 × 3 Gaussian convolution kernel through the Gaussian function for smoothing filtering of the neighboring pixels of the picture pixel.

The Gaussian convolution kernel is used to calculate the weighted average of each pixel in a certain neighborhood:(1)$$H_{i,j}=\frac{1}{2\pi \sigma^{2}}e^{-\frac{{\left(i-k-1\right)}^{2}+{\left(j-k-1\right)}^{2}}{2\sigma^{2}}}$$
where parameter *σ* represents the variance, and *k* represents the dimension of the convolution kernel matrix.

As an effective method of detecting moving objects, the optical flow method has been widely used in behavior recognition. Traditional optical flow estimation methods are divided into the dense optical flow and sparse optical flow. The sparse optical flow based on Lucas–Kanade has made certain progress in the research of fish movement detection and tracking [21]. However, both the detection accuracy and detection speed restrict the application in behavior recognition. Recently, the optical flow estimation method based on deep learning has shown excellent performance in terms of accuracy and speed [22].

In this work, the FlowNet2 network based on deep learning was applied to fish motion optical flow estimation [23], which provided motion feature information for behavior recognition. In particular, the behavioral characteristics of fish are mainly represented by the state of movement speed and direction. To this end, the optical flow videos were generated by FlowNet2 for motion estimation on the RGB videos. As shown in Figure 4, the optical flow image highlights the movement characteristics of the fish, and the movement direction is represented by the HSV image. The purpose of our method was to detect the movement of multiple objects in the RGB image through optical flow. To this end, we improved the original C3D model into a dual-stream network in which RGB video and optical flow video are simultaneously input. Feature fusion images contain appearance features and motion features, which provide more information about the fish movement.

FlowNet2 is an optical flow prediction method based on CNN, which models optical flow prediction as a supervised deep-learning problem:(2)W=CNN(θ,image1,image2)
where W is the predicted optical flow, the parameter θ is the parameter to be learned in the CNN, and image1 and image2 are the images to be input.

### 3.3. Spatiotemporal Feature Extraction

Recently, one possible solution for automatized analysis of behavior has been developed that trains a deep learning model on larger datasets for the purpose of predicting behavior types. For example, the 3D convolutional neural networks have been able to successfully applied to the recognition of human behavior, as they can extract the spatiotemporal features of moving objects between consecutive frames [20]. Inspired by human behavior recognition, we proposed a DSC3D network to extract spatiotemporal features of fish RGB videos and motion features in flow videos, respectively, as shown in Figure 5.

The original C3D network structure consists of eight convolutional layers, five pooling layers, two fully connected layers, and one softmax layer. The optimal convolution kernel size used by the eight convolution layers is 3 × 3 × 3. The numbers of convolution kernels used in each convolution layer are 64, 128, 256, 256, 512, 512, 512, and 512, respectively. In order to retain more temporal information in the initial stage of the network, the kernel size used by the first pooling layer is 1 × 2 × 2. After the convolution and pooling layer, a 4096-dim vector is generated and mapped to the predicted label after the softmax layer. The video clips are considered with a size of C × T × W × H, where C is the number of channels, T is the length of consecutive frames, W is the width of the image, and H is the height of the image. To unify the video format, we select consecutive frames with a length of 16 as the input of the model. The image size is adjusted to 1280 × 720 and then cropped to a size of 112 × 112. Since we converted the optical flow prediction result to the HSV color space, the channels of RGB video and Flow video are both 3.

When the quantity of training data is large, training deep neural network parameters from scratch is a time-consuming task. The transfer learning method was used to fine-tune the parameters of the pre-trained model trained on the UCF-101 data set (including 101 categories and a total of 13,320 videos). The number of parameters in our model was the same as the number of parameters in the original model. In each iteration, the batch size of the shuffled data input to the network was set to 16. Our network used the stochastic gradient descent method for training with a learning rate (lr) of 0.00005, a momentum of 0.9, and a weight attenuation of 0.01.

### 3.4. Image Fusion Scheme

Image fusion methods are designed to combine multiple images into a fused image, which further makes the image more informative [24]. Image fusion methods are divided into data-level fusion, feature-level fusion, and decision-level fusion. As far as our dataset is concerned, it is not necessary for data fusion due to the inaccurate registration of RGB images and flow images. The decision-level fusion requires multiple classifiers to perform a weighted average of results, such as voting and blending methods, but its results depend on the model results with higher weights. One major benefit of feature fusion is that it improves the performance of target detection and recognition by perfectly fusing feature information such as multi-level features, spatial features, and temporal features.

The feature fusion method of adding was used for image fusion, which is obviously helpful for image classification due to the increase in the amount of information on the feature maps. Feature fusion methods are divided into two types—namely, serial feature fusion (concat) and parallel feature fusion (add). The concat feature fusion method is realized by increasing the number of feature maps, while the add method superimpose the information of the feature maps and the number of channels remains unchanged. Therefore, the computational complexity of the concat method is much greater than that of the add method. We obtained the spatiotemporal features in the RGB image and the motion features in the optical flow image in the feature extraction stage, and then the two features were fused using the max pooling.

### 3.5. Experimental Evaluation Index

According to [16], accuracy, precision, recall, and specificity are used to evaluate the performance of the model. Accuracy is a measure of the accuracy of recognition of all behaviors, and it is the ratio of correct recognition samples to all samples. For a specific behavior, the accuracy is the ratio of correctly classified behavior samples to all predicted behavior samples. Recall rate is the ratio of correctly classified samples to the number of samples that belong to this behavior. Specificity is the proportion of genuinely negative samples among the negative results of the test. These four indicators are calculated as follows:(3)$$\mathrm{Accuracy}=\frac{TP+TN}{TP+FP+TN+FN}\times 100\%$$
(4)$$\mathrm{Precision}=\frac{TP}{TP+FP}\times 100\%$$
(5)$$\mathrm{Recall}=\frac{TP}{TP+FN}\times 100\%$$
(6)$$\mathrm{Specificity}=\frac{TN}{FP+TN}\times 100\%$$
where true positives (*TP*) indicate that the positive class was judged as a positive class, false positives (*FP*) indicate that the negative class was judged as a positive class, false negatives (*FN*) indicate that the positive class was judged as a negative class, and true negatives (*TN*) indicate that the negative class was judged to be the positive class.

## 4. Experimental Results

### 4.1. Result Analysis

#### 4.1.1. Comparative Analysis of Experiments

The average accuracy rate of the proposed algorithm for distinguishing the five behavior states of the fish school reaches 95.79%. As shown in Figure 6, the accuracy and loss curves of model training and verification tend to be in a stable close-companion state at the 20 epoch, which indicates that the model training results are robust and suitable for recirculating aquaculture systems. Our dataset includes the activity states of five fish schools that are identified by specific behaviors. Table 1 shows the description of each behavior. More details of the dataset are introduced in Section 2. For better experimental performance and model evaluation, we divide the dataset into 70% training set, 20% validation set, and 10% test set.

The results of precision, recall, specificity, and accuracy of the model on five different behaviors of fish are shown in Table 3. In addition to the misjudgment of a small number of samples, our proposed method realizes the accurate identification of fish behavior. To verify the effectiveness of the method, we selected 642 fish behavior videos as the test set. For example, the recognition accuracy of hypoxia and frightening behavior is low, achieving 96.9% and 97.2%, respectively. The confusion matrix of five behaviors is shown in Figure 7, including feeding, hypoxia, frightening, hypothermia, and normal state. From Figure 7, we derive the number of correctly predicted behaviors described on the diagonal line. The dark color indicates high confidence in prediction, while the light color shows low confidence in behavior misrecognition. This further shows the feasibility of our method for the accurate recognition of fish behavior.

We compared and analyzed the current main deep learning methods for fish behavior recognition on our dataset, including LeNet5 [16], 3D residual networks [25], and convolutional neural networks–long short term memory (CNN-LSTM) [17], as shown in Table 4. LeNet5 is a CNN-based method, which was used to evaluate the feeding intensity and appetite of fish by extracting image features of feeding behavior. In addition, 3D-ResNet is a deep convolutional neural network that extracts information in the temporal and spatial dimensions of the video and has been used to identify fish feeding and building behaviors. The CNN-LSTM architecture combines the feature extraction of CNN and the learning ability of recurrent neural network (RNN) for time series, which has been used for animal behavior recognition. Inspired by the dual-stream network, we directly extracted the spatiotemporal features in the RGB dataset and the motion features in the optical flow dataset through the C3D network and realized feature fusion in the last layer of feature extraction. The results show that the average accuracy of the algorithm is 2.46% higher than that of CNN-LSTM and 6.97% higher than that of the classic 3D residual networks. We found that the 3D convolutional network based on feature fusion as a feature extractor is more conducive to fish behavior classification.

#### 4.1.2. Comparison of Spatiotemporal Feature Extraction

In this section, different feature extraction networks were compared and analyzed, as shown in Table 5. Input resolution represents the size of the input network image, Params represents the model parameters of the neural network, and MACs represents the numbers of multiply-accumulate operations required by the algorithm. The results show that the C3D network achieves the best accuracy with moderate Params and MACs. More broadly, even if the C3D network is only used as a feature extractor without feature fusion, it still achieves an accuracy of 92.8%.

We compared and analyzed the I3D network and our method, both of which are dual-stream networks that take RGB videos and optical flow videos as input [26]. Although the improved network based on C3D has higher parameters and model complexity than I3D, our method achieves the highest behavior recognition accuracy rate of 95.79%. This shows that as a stable, simple, and efficient model, C3D is more capable of gathering the target, scene, and motion information in the video. Furthermore, I3D simply averages the results after training the network separately on RGB videos and optical flow videos, while our method considers the fusion of spatiotemporal features and motion features for accurate recognition of actions.

#### 4.1.3. Comparative Analysis of Feature Fusion

In this work, we attempted to further improve performance through feature fusion, and the fusion result is shown in Figure 8. To emphasize the importance of feature fusion, we also trained on the original RGB and optical flow data, respectively. It is evident that the training result of feature fusion is 2.99% higher than the single RGB data and 34.89% higher than the optical flow data. This shows that the fusion of spatiotemporal features and motion features can more effectively improve the performance of the model for detecting fish motion information.

We compared the effects of the early fusion and the late fusion on the model recognition performance by adding fusion features after Conv1 and Conv5b, respectively. As shown in Table 6, the results show that the recognition accuracy of high-level feature fusion is 3.89% higher than that of low-level fusion, and the training loss is reduced by 0.116. This indicates that the high-level features with strong semantic information further improve the detection ability of the region of interest in the image, while the feature image is low in resolution and poor in the perception of details. In addition, our method also adds more spatial motion information to the high-level features, which improves the efficiency of feature fusion.

#### 4.1.4. Visualization of Feature Map

The different motion patterns of the 3D CNN features of the five behavior sequences are shown in Figure 9. The feature visualization results help us understand the temporal and spatial regions that the model focuses on and display them in the form of heat maps. From Figure 9, the visualization results show the feature maps extracted by our method on the five behaviors, and the regions of interest in the feature maps corresponding to each behavior are disparate. The reason for this difference is that the fish exhibit very vigorous activities after being exposed to food and other external stimuli, while the fish exhibit an inactive state under conditions such as hypoxia and hypothermia. The similarities between fish behaviors are inevitable, while computer vision technology could still accurately distinguish the subtle changes in the temporal and spatial regions of interest. The experimental results show that the CNN feature map effectively distinguishes the difference in fish movement patterns, which is more conducive to the result of behavior classification.

### 4.2. Discussion

#### 4.2.1. The Basis of Fish Behavior Recognition Based on Video Analysis

The purpose of our research was to intelligently recognize five behaviors of fish, including feeding, hypoxia, hypothermia, frightening, and normal behavior. The recognition of typical behaviors is useful to understand the real activities of fish, which directly helps to discover the health status and diseases of fish. For example, video-based behavior analysis can also accurately quantify fish activity that is closely related to the health of fish and provides help for the diagnosis of an environmental stress response. In related work, researchers mostly use acceleration sensors and other equipment implanted in fish skin and body, combined with machine learning methods to realize fish behavior recognition. However, this method requires high accuracy of the acceleration sensor underwater, and this operation is likely to cause stress to the fish. Furthermore, the method based on acceleration sensors still has difficulty in identifying more complicated fish behaviors. Therefore, the advantages and potentials of fish behavior recognition based on video analysis are incomparable.

The automatic recognition of fish behavior states was realized by deep learning methods, and the main behaviors of fish in aquaculture were collected. However, fish behavior recognition based on video analysis is still a challenging task under conditions of insufficient light. Researchers using near-infrared machine vision to quantify fish feeding behavior creates a new solution for behavior recognition under insufficient light conditions [7,27]. As described in Section 4.1.1, our proposed method realized the behavior recognition of fish under different environmental conditions, which lays the foundation for more complex behavior recognition of fish.

#### 4.2.2. The Problem of Identification of Similar Behaviors

The recognition method based on deep learning was implemented by extracting rich feature information from video clips. We highlighted the movement behavior of fish in the feature image by fusing spatiotemporal features and motion features. The results show that our proposed method further improves the accuracy of behavior recognition and meets the needs of behavior classification in aquaculture.

However, fish behaviors are susceptible to external environmental stimuli, showing a series of complex behavior states. Therefore, many similar behavior fragments lead to the misjudgment of the model, as shown in Figure 9. The most significant is that both the feeding and frightening behaviors stimulated by the external environment exhibit the aggregation state of the fish school. This behavioral change involves the rapid movement of the fish to a certain position. The similarity of behavioral activity is the reason for the misjudgment of feeding behavior and frightening behavior. In addition to floating at the inlet of the circulating water, fish also exhibit slow swimming behavior under hypoxia. This inactive behavior often occurs under normal conditions, and its appearance is similar to hypoxic behavior. Therefore, the research should focus on how to extract deep feature information and accurately identify behaviors in different environments.

#### 4.2.3. Other Behavior Recognition

In our study, only five basic fish behaviors were identified, including feeding, hypoxia, hypothermia, frightening, and normal behavior. However, fish also exhibit higher-level behaviors such as fish reproduction and fighting in real farming. Reproductive behavior is an essential behavior to optimize the breeding level in aquaculture. We focused on the study of fish movement behavior, while reproductive behavior is difficult to distinguish from normal behavior. Fighting behavior is a competitive behavior in which two or more fishes compete for resources, but it is not easy to collect behavioral data. Furthermore, a general network is difficult to recognize multiple behaviors at the same time, which will reduce the confidence of the network prediction result. Therefore, future research will consider more types of behaviors and further improve the ability to predict multi-classification tasks. In the foreseeable future, the fusion of machine vision methods and acoustic imaging methods will realize the intelligent recognition of fish behavior in aquaculture.

## 5. Conclusions

In this paper, we proposed a DSC3D network based on deep learning for fish behavior recognition with spatiotemporal information and motion information. This research further expands the idea of fish schools’ behavior recognition through video analysis and specifically focused on animal group behavior. Our proposed method used C3D as a feature extractor to extract spatiotemporal features of RGB videos and motion features of optical flow videos, and feature maps fused with depth feature information were used to classify fish behaviors. This method accurately identifies the five behavioral states of the fish school, including feeding, hypoxia, hypothermia, frightening, and normal behavior. The average accuracy of this algorithm for fish behavior recognition has reached 95.79%, which significantly proves the effectiveness of our strategy on the test dataset. The RGB camera used in our method can be operated in real farming, which is helpful to promoting daily management work. More broadly, we also discussed the challenges of understanding and recognizing fish behavior to improve the efficiency of farming and animal welfare. In the future, our work needs to consider identifying and tracking the behavior of each individual to understand more complex fish behaviors.

## Figures and Tables

**Figure 1 animals-11-02774-f001:**
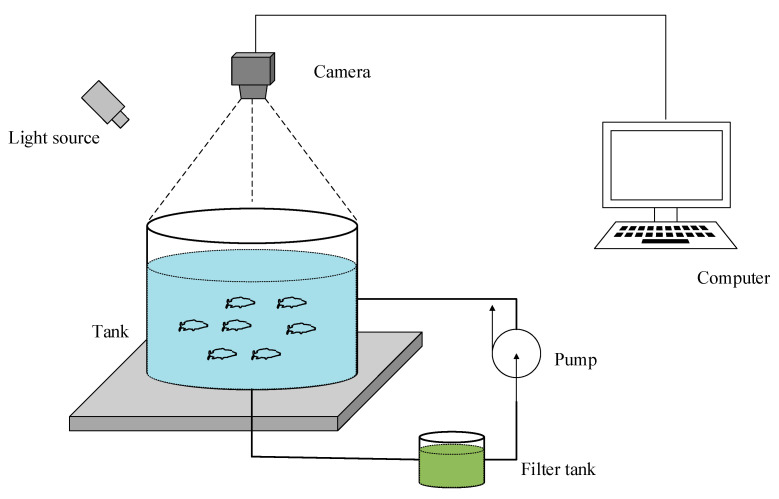
The layout of the experimental scene.

**Figure 2 animals-11-02774-f002:**
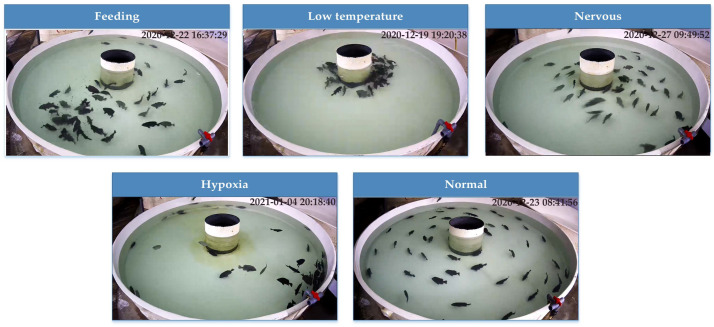
Keyframes of the fish behavior dataset.

**Figure 3 animals-11-02774-f003:**
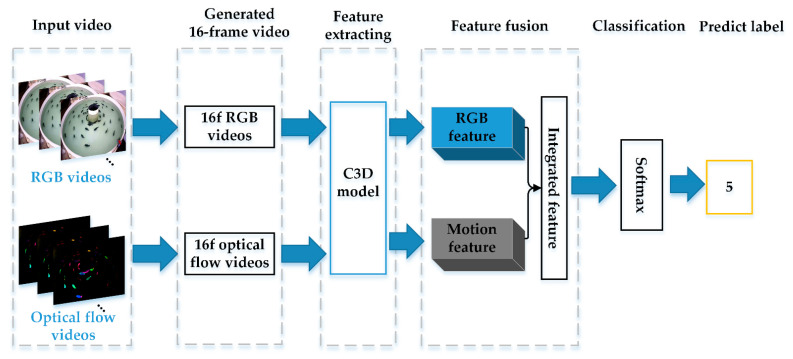
Overview of our proposed deep learning-based fish behavior recognition method.

**Figure 4 animals-11-02774-f004:**
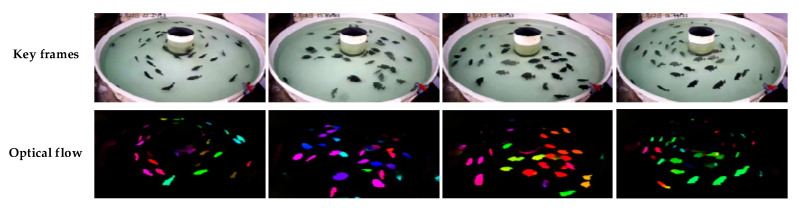
Keyframes and optical flow estimation in a video sequence.

**Figure 5 animals-11-02774-f005:**
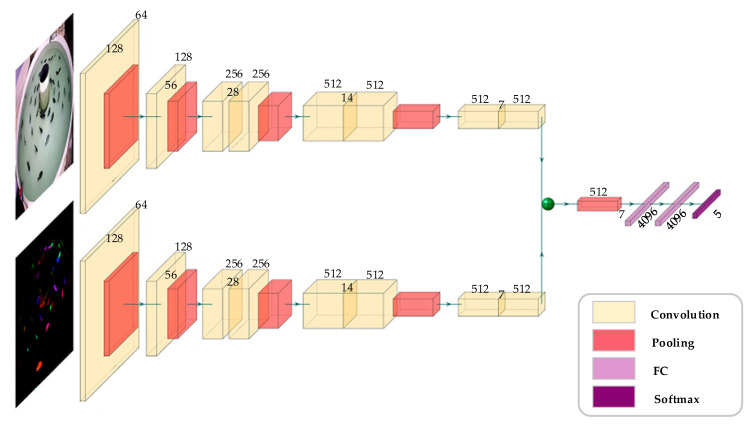
The network architecture of the DSC3D model with two-stream input.

**Figure 6 animals-11-02774-f006:**
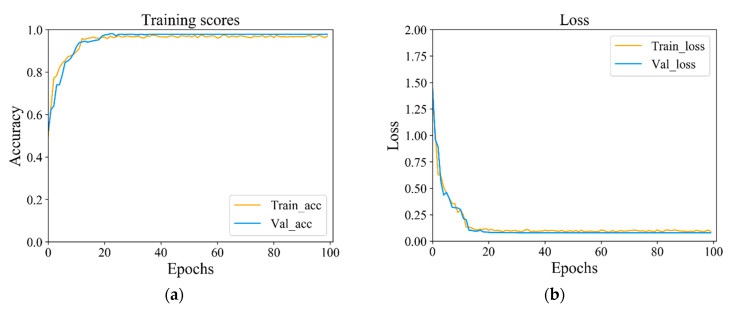
Accuracy and loss curves of training and validation sets: (**a**) the accuracy of training and validation; (**b**) loss curves for training and validation sets.

**Figure 7 animals-11-02774-f007:**
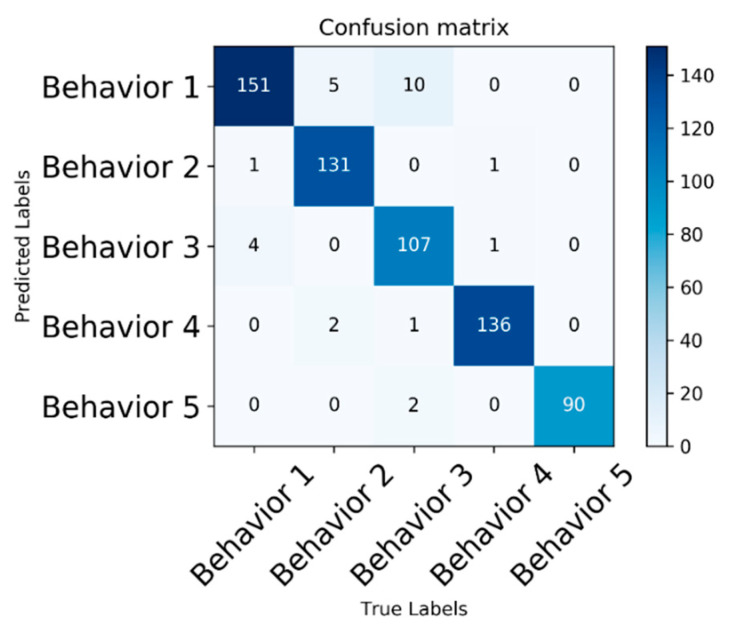
Confusion matrix results. Behaviors 1–5 represent hypoxia, feeding, frightening, hypothermia, and normal behaviors, respectively.

**Figure 8 animals-11-02774-f008:**
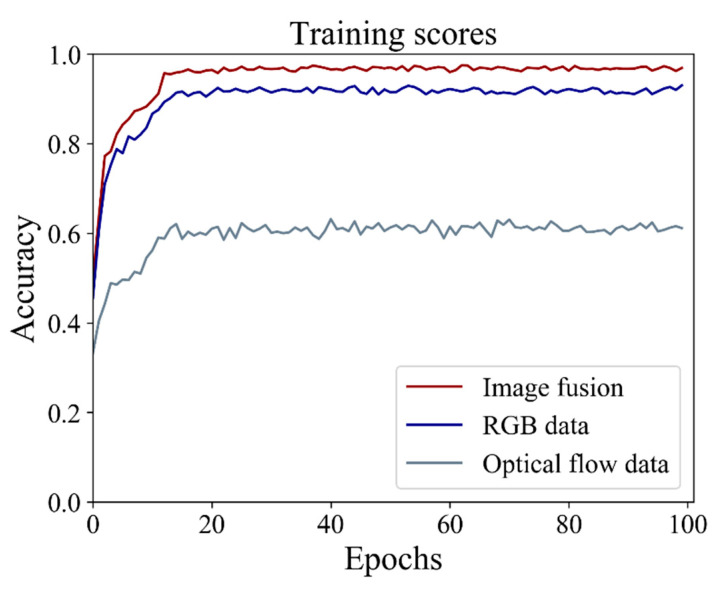
The result of feature fusion and the training result on the original RGB and optical flow data.

**Figure 9 animals-11-02774-f009:**
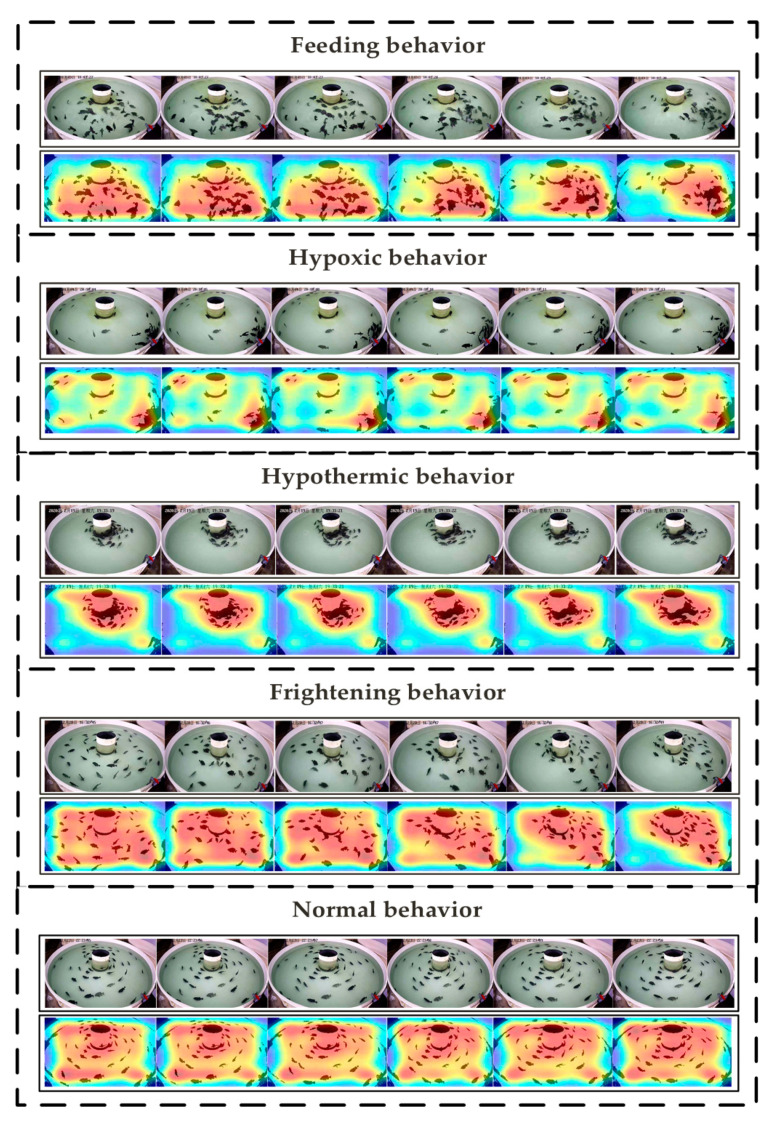
Visualization of 3D CNN features of five behaviors.

**Table 1 animals-11-02774-t001:** Description of fish behavior based on video analysis.

Types of Behavior	Description of Behavior
Feeding	Eating bait in groups.
Hypoxia	Floating heads and gathering at the water inlet of the fish tank.
Hypothermia	Sinking to the bottom of the pool, inactive.
Frightening	Swimming quickly after feeling external stimuli (sound or light).
Normal	Swimming coordinately in the entire tank.

**Table 2 animals-11-02774-t002:** Statistics of fish behaviors in our dataset.

**Category**	**Number of Videos**	**Average Video Duration(s)**	**Frame Rate/fps**	**Number of Key Frames**
Feeding	690	8	25	138,000
Hypoxia	780	8	25	156,000
Hypothermia	682	8	25	136,400
Frightening	596	8	25	119,200
Normal	450	8	25	90,000
Total	3198	8	25	639,600

**Table 3 animals-11-02774-t003:** The precision, recall, specificity, and accuracy of different behaviors of fish.

**Behaviors**	**Precision**	**Recall**	**Specificity**	**Accuracy**
Feeding	98.5%	94.9%	99.6%	98.6%
Hypoxia	91%	96.8%	96.9%	96.9%
Hypothermia	97.8%	98.6%	99.4%	99.2%
Frightening	95.5%	89.2%	99%	97.2%
Normal	97.8%	100%	99.6%	99.7%

**Table 4 animals-11-02774-t004:** Comparison of our method with other methods in our dataset.

Methods	Accuracy
LeNet5	87.06%
3D residual networks	88.82%
CNN-LSTM	93.33%
Proposed framework	95.79%

**Table 5 animals-11-02774-t005:** Comparison of different feature extraction networks.

**Model**	**Input Resolution**	**Params (M)**	**MACs (G)**	**Test (%)**
I3D	224 × 224	11.72	51.84	74.40%
3D-Resnet101	112 × 112	117.96	228.54	64.7%
C3D	112 × 112	74.40	71.84	92.8%

**Table 6 animals-11-02774-t006:** Comparison of feature fusion in different periods.

**Periods**	**Test Accuracy**	**Training Loss**
Early fusion	91.9%	0.2076
Late fusion	95.79%	0.0916

## Data Availability

The dataset can be obtained by contacting the corresponding author.

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
