# Peer review of "Automatic Recognition of Fish Behavior with a Fusion of RGB and Optical Flow Data Based on Deep Learning"

_animals, 2021, doi:10.3390/ani11102774_

Round 1
Reviewer 1 Report
The paper considers automatic recognition of five fish’s behaviors: feeding, hypoxia, hypothermia, frightening, and friendly behavior, using computer vision and deep learning. The proposed solution is based on a fusion of RGB and optical flow.
The paper is interesting, correctly organized and well written. Results of exhaustive experiments comparing known and new approaches are convincing.
Minor remarks:
Line 63: can be capture,
Line 118: 2.1. fish,
Line 247: Flownet2 whereas in line 236 is FlowNet2,
Line 283: what do you mean under “kinetic energy” ? I expected momentum,
Is your dataset publicly available?
Reviewer 2 Report
The present work by Wang and colleagues (ID animals-1352601) proposed a new model for behavioral classification based on the processing of video clips from a 3D convolutional neural network. The model uses RGB and optical flow video clips as input to classify behavioral conditions of fish schools. Based on class selection by fish-farming experts, the model was trained on classifying 5 different types of behavior (supervised learning): feeding, hypoxia, hypothermia, frightening, and friendly behavior. Results showed a higher performance of the proposed model (~96%) compared to the previous algorithms, providing a potential tool for automatically monitoring fish conditions in order to reduce human intervention while increase animal welfare.
For my concern, I am not a fully expert on Deep Learning. However, the authors provide several details on the model in a fluent style that allows the reader to easily understand their work. The study is well conducted, and the method is appropriate. I personally consider the present manuscript as a suitable contribution to the field of animal farming and welfare, as well as an interesting point for other disciplines (e.g. investigate the effect of pharmacological or chemical drugs on fish behavior and cognition). I have some minor comments for the authors.
Abstract (lines 24-28) & Introduction (lines 56-64)
Authors reported biotelemetry methods as the common approach for identifying fish behavior. This sentence is true to some specific fields, such as animal welfare monitoring or farming, however, other approaches have been used for investigating behavior and cognitive style changes. For example, several algorithms for video-tracking (e.g., Ethovision, https://www.noldus.com/ethovision-xt?gclid=Cj0KCQjwsZKJBhC0ARIsAJ96n3VQ2F-39sGue-miGhIjJUCPV1rQr9cMscwoCWtK8x60MP1iNSIK6AsaAreZEALw_wcB ) or automated apparatus (e.g. Zantkis https://zantiks.com/ ) have been used for automatically modeling fish behavior, but limiting to one or few subjects per time. My suggestion is to highlight the differences between biotelemetry methods and other techniques in terms of the trade-off between the number of studied subjects and breeding conditions. I consider the suggested method by Wang and colleagues a relevant milestone to fill the gap, i.e. collecting real-time data from multiple subjects for investigating their behavior in a natural/semi-natural condition.
Model comparison (lines 352-367).
I have some questions for the authors regarding the comparison between the novel model and the current models used for fish behavior recognition (i.e., LeNet5, 3D Residual Networks, Dual-Stream Recurrent Network).
I did not understand how authors have conducted model comparisons. In particular, I was not sure whether models were compared on their accuracy based on their own dataset or based on the new dataset proposed by Wang and colleagues.
In case the comparison has been conducted considering the model’s performance from their images dataset, different accuracy may be attributed to the type/quality of images rather than the limited capacity of the model to feature extraction.
In case the performance has been compared considering the same dataset as input, differences may be attributed to the pre-trained weights or to the final classification model. However, a fine-tuning of the previous models by pruning the classifier task and trained a new classifier as suggested by Wang and colleagues may be more suitable for model comparison.
In case the performance was evaluated only for the quality of feature extraction methods, the adopted dataset for compared the approximate matrix factorization may influences model performance.
I would ask the authors to provide more description on how the models’ comparison has been performed.
